# A Space Fractional Uphill Dispersion in Traffic Flow Model with Solutions by the Trial Equation Method

Rfaat Moner Soliby * and Siti Suhana Jamaian

Department of Mathematics and Statistics, Faculty of Applied Sciences and Technology, Universiti Tun Hussein Onn Malaysia, Pagoh Educational Hub, Johor Bahru 84600, Malaysia
* Correspondence: hw200021@siswa.uthm.edu.my

**Abstract:** This paper has two main objectives. First, we modify the traffic flow model by introducing the uphill dispersion that derives from the fact that, in peak hours, drivers tend to travel from low to high density regions. This means that the proposed model recovers wrong-way travel and is free from advected discontinuity. Second, in order to describe the anomalous transport behavior, we fractalize the proposed model to include dynamics with the fractional in space. As a result of adopting the fractional Fick's law, several moving jam waves are presented which elucidate the non-homogeneity of driving styles. Then, the *GFFD* fractional derivative and the trail equation method are applied and for some special cases solutions are simulated which could help transportation engineers to understand traffic behavior and thus make appropriate decisions when constructing a traffic signals network.

**Keywords:** anomalous transport behavior; LWR model; trial method

## 1. Introduction

In recent years, traffic jams have begun to intensify with each passing day and have become a major social problem in many cities. This will result in more wasted fuel, more carbon dioxide emissions, longer destination times and less economic productivity. There are multiple factors causing traffic jams, poor traffic infrastructure is perhaps the most critical one. Thus, in constructing traffic facilities such as signals, junctions or bridges, transportation engineers are required to make the right decisions, otherwise congestion is inevitable. Hence, many traffic flow models have been proposed to describe the traffic behavior. In fact, the Lighthill–Whitham–Richards model [1,2], denoted by LWR, is considered the foundation stone of all continuum vehicular models. LWR is a scalar hyperbolic conservation law and thus admits the moving of discontinuous travel upstream in the density profile; this is a sharp transition from one density state to another and which leads to unbounded acceleration or deceleration [3,4]. Moreover, the classical LWR model fails to accurately characterize various complex phenomena, such as non-equilibrium traffic flow and the anisotropic nature of traffic flow [5]. Due to the presence of many flaws, substantial research in traffic flow modeling has developed from different aspects to study the dynamic evolution process of traffic flow. For instance, Payne [6] and, independently, Whitham [7] have proposed a 2 × 2 model by adding a dynamic velocity equation based on car following theory. This quasi-linear hyperbolic system was originally derived from the similarity between traffic flow and fluid flow and it includes two new terms, the relaxation term to verify that the model tends to relax to an equilibrium velocity and the anticipation term which accounts for drivers' reactions to downstream traffic. However, the assumption that all drivers engage in similar behavior leads to unrealistic results [8]. On different approaches, many higher-order models were proposed wherein the dispersion term was added in order to smooth the shock [9,10]. Inspired by the isothermal law, Aw and Rascle [11] introduced a new second-order model in which velocity is a monotonically

increasing function of density. Yet, with high density the model admits infinite acceleration or deceleration which is impractical [12].

However, Daganzo [13] criticized all of the higher-order models because they may require a violation of the anisotropic nature of the traffic. Another major drawback of this approach is that the speed and density is merely a fixed curve which means the bivariate relation between traffic variables is always assumed to be in an equilibrium state [14]. However, traffic is generally observed in non-equilibrium [15], so that the traffic dynamics should not be restricted to a fixed curve in a density–space plane. Physically speaking, drivers' behaviors could vary slightly in response to a jam, so they may alter their speed based on many factors such as gender, experience, attitude, age, driving conditions, etc. Such a deviation from the equilibrium state could be represented by anomalous dispersion [16].

More recently, in comparison with classical calculus, fractional calculus has proved to be an effective tool in modeling various phenomena in different areas of science with greater degree of accuracy. Therefore, much attention has been paid to investigating the effects of the fractional order on complex systems such as in a pandemic [17], medicine [18], economics [19], engineering [20], optimal control [21] and physics, for instance with the thermostat process [22], or the anomalous dispersion process [23].

Over the past several decades, anomalous dispersion has been observed in a wide diversity of systems. Unlike the classical dispersion represented by Fick's law, anomalous dispersion is considered as a non-Fickian transport process which describes the behavior of early arrivals, something which cannot be captured by the conventional Fick's law. Furthermore, by using Fick's law along with the traditional conservation of mass equation, we implicitly admit that the flux through a combined area is proportional to the density gradient, which is to say that it is a linear relation [24], while the fractional non-Fickian flux exhibits a nonlinear relationship according to the space's fractional dispersion. The physical interpretation is that the change in flux is no more linear and follows the power law due to the effect of the fractional order, thus our proposed model assumes that drivers respond with a delay to changes in traffic conditions that is nonequilibrium [25].

Without doubt, solving fractional differential equations is difficult and, most of the time, these solutions cannot be found analytically, or even numerically. Yet, due to their prevalence in a wide range of applications, several successful attempts have been made to find an exact, accurate and reliable solution such as the Chebyshev integral operational matrix method [26], the Laplace transform method [27], the variational iteration method [28], the Galerkin method [29], the Jacobi elliptic function expansion method [30] and the trial equation method [31].

The aim of this paper is to build a modified vehicular fractional LWR model to remove the possibility of cars going backward, and to explain various traffic behaviors which are of significant importance to transportation engineers. The structure of this article is schematized as follows. In Section 2, we introduce the *GFFD* fractional derivative and the trial equation method. In Section 3 we reform the model by modifying the dispersion term. In Section 4, solutions are obtained for the proposed model. In Section 5, a traffic simulation is performed. Finally, we offer conclusions in Section 6.

## 2. Proposed Methodology

In this section, we introduce the definition of *GFFD* fractional derivative and some basic properties. Furthermore, we describe the main steps of the trial method.

### 2.1. The GFFD Fractional Derivative

**Definition 1.** *Let $f : [0, \infty) \to \mathbb{R}$. Then, the GFFD fractional derivative of f of order $0 < \alpha \leq 1$, is defined as,*

$$^{GFFD}D_x^\alpha f(x) = \frac{\partial^\alpha f(x)}{\partial x^\alpha} = \lim_{\varepsilon \to 0} \frac{f\left(x + \frac{\Gamma(\beta)}{\Gamma(\beta+1-\alpha)}\varepsilon x^{1-\alpha}\right) - f(x)}{\varepsilon},$$

$$\text{for } \beta \in (-1,0) \cup (0,+\infty).$$

**Theorem 1.** *Let f, g be $\alpha-$ differentiable functions at a point $x \geq 0$, then one has the following,*

$$\frac{\partial^\alpha}{\partial x^\alpha}(af \mp g)(x) = a\frac{\partial^\alpha}{\partial x^\alpha}f(x) \mp \frac{\partial^\alpha}{\partial x^\alpha}g(x), \forall a \in \mathbb{R} \tag{1}$$

$$\frac{\partial^\alpha}{\partial x^\alpha}(f \cdot g)(x) = g(x)\frac{\partial^\alpha}{\partial x^\alpha}f(x) + f(x)\frac{\partial^\alpha}{\partial x^\alpha}g(x) \tag{2}$$

$$\frac{\partial^\alpha}{\partial x^\alpha}x^\beta = \frac{\Gamma(\beta+1)}{\Gamma(\beta+1-\alpha)}x^{\beta-\alpha} \tag{3}$$

$$\frac{\partial^\alpha}{\partial x^\alpha}\lambda = 0, \forall \lambda \in \mathbb{R} \tag{4}$$

$$\frac{\partial^\alpha}{\partial x^\alpha}f(x) = \frac{\Gamma(\beta)}{\Gamma(\beta+1-\alpha)}x^{1-\alpha}\frac{df}{dx}(x) \tag{5}$$

Additional details and proofs are provided in [32,33].

*2.2. Outline of the Trial Equation Method*

Consider the following fractional differential equation [34],

$$F\left(\rho, \rho_t, \rho_x, D_x^\alpha\rho, D_x^{2\alpha}\rho, \dots\right) = 0. \tag{6}$$

- Step 1. Using the fractional transformation,

$$\rho(x,t) = \rho(\xi), \xi = k\frac{\Gamma(\beta+1-\alpha)}{\alpha \cdot \Gamma(\beta)}x^\alpha - \mu t, \tag{7}$$

where $k, \mu$ are nonzero constant. Equation (6) is then converted to a nonlinear ordinary differential equation as,

$$F_1\left(\rho, \rho', \rho'', \dots\right) = 0, \tag{8}$$

where $F_1$ is a polynomial of $\rho$ and its derivatives and the notation $\prime$ denotes the derivative with respect to $\xi$.

- Step 2. Suppose the trial equation is of the form,

$$\rho' = \frac{G(\rho)}{H(\rho)} = \frac{\sum\limits_{i=0}^{N} a_i\rho^i}{\sum\limits_{j=0}^{M} b_j\rho^j}, \tag{9}$$

where $a_i(i = 0, 1, \dots, N)$ and $b_j(j = 0, 1, \dots, M)$ are all constant and $a_N, b_M \neq 0$. $N$ and $M$ are positive integers which can be determined by balancing the linear term of the highest order with the highest order of nonlinear term, whereas $G$ and $H$ are polynomials of $\rho$. Placing Equation (9) into Equation (8) precedes an equation of polynomial $\Psi$ of $\rho$ as follows,

$$\Psi(\rho) = \theta_r \rho + \dots + \theta_1 \rho + \theta_0 \rho = 0. \tag{10}$$

Based on the balance principle, one can find a relation between $N$ and $M$. So, multiple solutions can be achieved.

- Step 3. Setting the coefficients $\theta_l (l = 0, 1, \ldots, r)$ to zero yields a system of algebraic equations concerning the unknowns $a_i$, $b_j$, $k$ and $\mu$ Then, we solve this system to determine the values of $a_0, a_1, \ldots, a_N$ and $b_0, b_1, \ldots, b_M$ with the help of symbolic computation software such as Maple 2021.
- Step 4. Rewrite Equation (9) in the classical integral form as,

$$(\xi - \lambda) = \int \frac{H(\rho)}{G(\rho)} d\rho , \tag{11}$$

where $\lambda$ is a constant to be determined later. Applying the complete discrimination system for the polynomial $G(\rho)$, we can know the number and multiplicities of the distinct real roots of polynomial $G(\rho)$. Finally, by solving the infinite integral Equation (11) the exact solutions of Equation (6) will be derived.

## 3. Problem Formulation

It is well known that the classical LWR fails to model many traffic features, such as the anisotropic property of the traffic flow or the heterogeneity of the driving styles. Motivated by the desire to collaborate between two fundamental concepts, namely the uphill dispersion and the fractional dispersion, a new model is constructed. The first approach is introduced in order to address the issue related to the anisotropic nature of traffic flow by mimicking the uphill dispersion. The next stage in the development is to upgrade the uphill dispersion into the fractional dispersion so that the model now admits the phenomena of anomalous dispersion which cannot be explained by the classical calculus. As a result the drivers can travel faster or slower based on the value of the fractional order, which explains the non-homogeneities of the traffic flow.

To begin, the classical LWR model is given by,

$$\frac{\partial \rho(x,t)}{\partial t} + \frac{\partial Q(x,t)}{\partial x} = 0, \qquad (x,t) \in \upsilon = [0,L) \times [0,\infty), \tag{12}$$

where $\rho(x,t)$ is traffic density, representing the number of vehicles per unit length, and $Q(x,t)$ is the flow rate. In fact, without the framework of Fick's law the concentration curve breaks and produces moving discontinuity, sometimes called shock [35], leading to a sudden change in a vehicles' velocity which finally causes an accident because drivers are assumed to reach different equilibrium velocities after a sudden change in the traffic state from low to high density [36], which implies infinite deceleration as shown in Figure 1a. To overcome this problem, higher-order models were suggested wherein the dispersion term was added to reflect drivers slowing down gradually when they see that the traffic density is increasing [6,7]. Mathematically, higher-order models indicate that the traffic flow on road is a function of both traffic density $\rho$ and its gradient $\rho_x$ given by Fick's first law. Thus, the total flow will be the sum of advected flux at a velocity $v$ and dispersive flux as,

$$Q_{Total}(\rho) = Q_{Advective} + Q_{Dispersive}, \tag{13}$$

using the hydrodynamic relation and Fick's first law, Equation (13) would be,

$$Q_{Total}(\rho) = \rho v - \delta \rho_x. \tag{14}$$

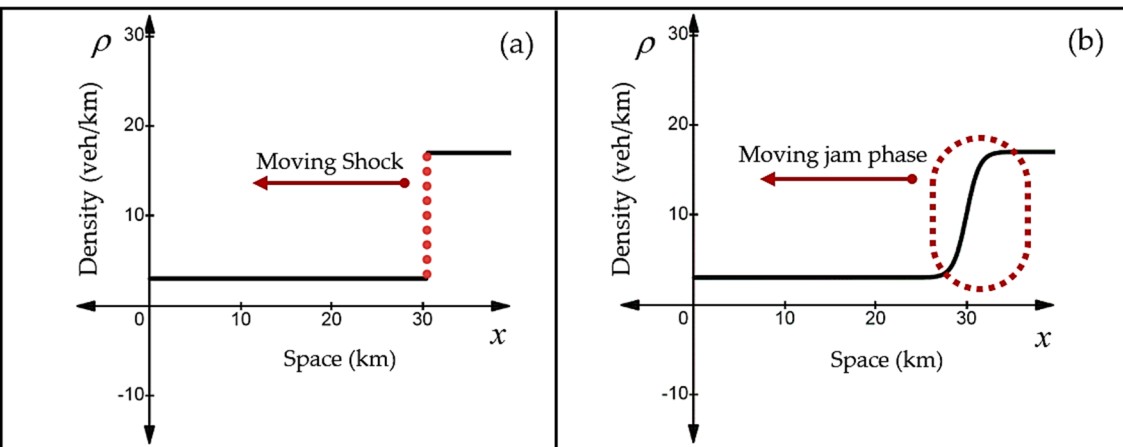

**Figure 1.** Moving jam: (**a**) Discontinuous moving jam due to infinite deceleration; (**b**) smooth moving jam phase due to gradual deceleration.

It is assumed and proved that the diffusivity $\delta$ is a non-negative constant. Substituting Equation (14) in Equation (12) yields,

$$\frac{\partial \rho}{\partial t} + \frac{\partial (\rho \cdot v)}{\partial x}\bigg|_{Advection} - \delta \frac{\partial^2 \rho}{\partial x^2}\bigg|_{Dispersion} = 0. \tag{15}$$

However, in a novel paper introduced by Daganzo 1988 [13], the author criticized these models and proved that, though these models are able to eliminate all the shocks and smooth the density profiles as depicted in Figure 1b, in doing so they violate the anisotropic nature of traffic flow, implying that the drivers in these models may travel backwards under some conditions while in real-life situation this is neither true nor authorized in most cities.

To clarify, in a moving jam phase as shown in Figure 1b, the density gradient $\rho_x$ is substantially large, which makes the total flow in Equation (14) negative and thus demonstrating that the model produces a 'wrong-way travel' that is negative flow and, hence, negative travel speed. In our study we modify the model by replacing the classical dispersion with an uphill dispersion in order to deal with this negative flow.

First, in 1855, Adolf Fick described Fick's law of diffusion as,

$$Q_{dispersive} = \delta\left(-\frac{\partial \rho}{\partial x}\right). \tag{16}$$

Originally, this law was derived from the fact that particles flow from regions of high density to regions of low density, which considered a negative change. Since flux is a positive quantity, the negative sign was added to Fick's law so that the flow ultimately becomes positive [37]. Put differently, the negative sign appears due to the opposite direction of particle flow and concentration gradient as depicted in Figure 2a. That is, the particle flows in the direction of decreasing solute concentration. Contrary to particles theory, traffic flow theory behaves in such a way that drivers in general tend to go from low density to high density as a daily routine when going to work, especially during the rush hour, such movement is called active transport because it is against the density gradient as shown in Figure 2b.

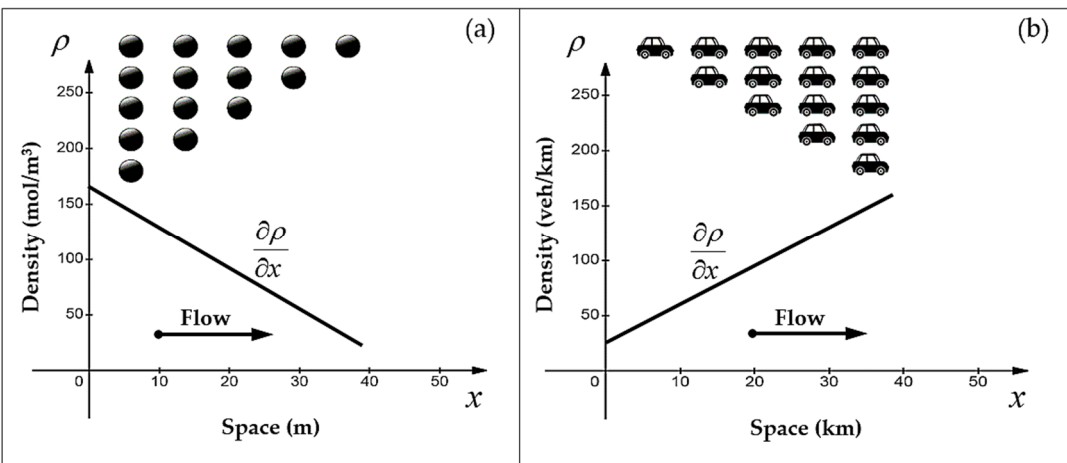

**Figure 2.** Downhill and uphill dispersion: (**a**) Particles disperse down the density gradient; (**b**) vehicles disperse against the density gradient.

In other words, vehicles spread in the direction of increasing rather than decreasing density. Therefore, the negative sign will be removed from Fick's law and the model now follows the so-called uphill dispersion. In fact, this kind of propagation is widely observed in nature such as when a system simulates a vapor–liquid phase transition [38]. Over and above that, this modification is not considered a violation of the second law of thermodynamics [39].

Under such conditions stated earlier, Equation (14) can be rewritten as,

$$Q_{Total}(\rho) = \rho v + \delta \rho_x, \tag{17}$$

substitute Equation (17) in Equation (12) we have,

$$\frac{\partial \rho}{\partial t} + \left.\frac{\partial(\rho \cdot v)}{\partial x}\right|_{Advection} + \left.\delta \frac{\partial^2 \rho}{\partial x^2}\right|_{Dispersion} = 0. \tag{18}$$

Second, as is widely known, nature is not always homogeneous or predictable so that it may change over time and/or space, thus research and experiments related to time- or space-dependence are of great importance. Indeed, a time- or/and space fractional model is considered an effective tool to describe the unpredicted and anomalous behavior of a complex system [40,41], because the fractional non-integer order acts as an additional parameter that offers flexibility in the simulation processes. Moreover, large numbers of experiments imply that the anomalous transport cannot be described by the classical calculus [42]. In fact, the classical Fick's law fails to describe the anomalous behavior, in that a fractional Fick's law is required. According to [43], if one describes the flux of particles as proportional to a fractional derivative, then the magnitude of the particle velocities is changeable. To put it concisely, using the fractional Fick's law means that the ratio of particles jump is no longer constant.

Therefore, the fractional LWR model is considered a generalization of the classical LWR model, as a result the density, velocity and the flux now satisfy the scaling law $x \sim x^\alpha$. Moreover, the fractional measure $dx^\alpha$ is utilized instead of the classical measure $dx$ and hence the travelled distance that occurs during a given $dt$ is highly affected by the scaling index $\alpha$. Consequently, the model accounts for anomalous features [44]. In analogy, some drivers may temporarily exceed their equilibrium values, so they speed a little, up or down. In fact, this anomalous behavior is a result of different driving personalities.

On that account, we replace the classical derivative in space by a fractional derivative in Equation (18), so,

$$\frac{\partial \rho}{\partial t} + {}^{GFFD}D_x^{\alpha}\ (\rho \cdot v)\Big|_{Advection} + \delta \cdot {}^{GFFD}D_x^{2\alpha}(\rho)\Big|_{Dispersion} = 0. \tag{19}$$

Using the Greenshield speed–density relationship [45],

$$v = v_m\left(1 - \frac{\rho}{\rho_m}\right), \tag{20}$$

where $v_m$ and $\rho_m$ are the maximum speed and density, respectively. Equation (19), is then written as,

$$\frac{\partial \rho}{\partial t} + {}^{GFFD}D_x^{\alpha}\ \left(v_m \cdot \rho - \frac{v_m}{\rho_m}\rho^2\right) + {}^{GFFD}D_x^{2\alpha}(\delta\rho) = 0. \tag{21}$$

Equation (21) represents the fractional uphill model. Obviously, as the anomalous dispersion exponent $\alpha$ approaches 1, the model tends toward the classical LWR where the flow is proportional to the first derivative rather than the fractional derivative.

## 4. Solutions

Use of the transformation Equation (7) and the properties (1–5) changes Equation (21) into the following partial differential equation,

$$-\mu\rho' + kv_m \cdot \rho' - \frac{kv_m}{\rho_m} \cdot 2\rho\rho' + \delta \cdot k^2\rho'' = 0 , \tag{22}$$

integrating Equation (22) once yields,

$$(-\mu + kv_m)\rho - \frac{kv_m}{\rho_m} \cdot \rho^2 + \delta \cdot k^2\rho' + C = 0, \tag{23}$$

where $C$ is the integration constant. Substituting Equation (9) into Equation (23) and balancing $\rho^2$ with $\rho'$ gives,

$$N - M = 2, \tag{24}$$

for simplicity, we set $N = 2$, so Equation (9) is reduced to,

$$\rho' = \frac{G(\rho)}{H(\rho)} = \frac{a_0 + a_1\rho + a_2\rho^2}{b_0}, \tag{25}$$

where $a_2$ and $b_0$ are non-zero constants. Substituting Equation (25) in Equation (23), collecting all terms with the same powers and equating each coefficient of the polynomials in Equation (10) to zero, then solving the over-determined algebraic equations by Maple, we can obtain the following results,

$$\left\{a_0 = -\frac{b_0 \cdot C}{\delta k^2}, a_1 = -\frac{b_0(kv_m - \mu)}{\delta k^2}, a_2 = \frac{b_0 \cdot v_m}{\delta k\rho_m}\right\} \tag{26}$$

as long $b_0$ is a free parameter, we set $b_0 = 1$. Substitute system (26) in Equation (11), we have,

$$\pm (\xi - \lambda) = \int \frac{1}{\frac{v_m}{\delta k \rho_m}\rho^2 - \frac{(kv_m - \mu)}{\delta k^2}\rho - \frac{C}{\delta k^2}}d\,\rho, \tag{27}$$

by complicated but standard computation we obtain,

$$\rho = \frac{\rho_m(kv_m - \mu)}{2kv_m} + \frac{\sqrt{\rho_m\left(4Ckv_m + \rho_m(kv_m - \mu)^2\right)}}{2kv_m}\tanh\left(0.5\sqrt{\frac{4Cv_m}{\delta^2k^3\rho_m} + \frac{(kv_m - \mu)^2}{\delta^2k^4}} \cdot \left(k\frac{\Gamma(\beta + 1 - \alpha)}{\alpha \cdot \Gamma(\beta)}x^{\alpha} - \mu t - \lambda\right)\right). \tag{28}$$

Equation (28) is a family of exact solutions for Equation (21).

## 5. Simulation

In this section, we consider a red-light signal in which the density from vehicles on the right reaches a maximum capacity at $\rho_r = 120$ veh/km, given that the mean length of a vehicle is 5 m and the safe separation distance between any two successive and stopping vehicles is about 3 m. Thus, the velocity will be zero due to the vehicles stopped at the red-light signal with zero speed, satisfying the Greenshield relation Equation (20). The density from the left is assumed to be very low, for example, $\rho_l = 20$ veh/km. Owing to the fact that, $\tanh(y) \in (-1, +1)$, $\forall y \in \mathbb{R}$, $\rho_{\max} = 120$ and $\rho_{\min} = 20$, Equation (28) is reformulated as the following system,

$$
\begin{cases}
\dfrac{\rho_m(kv_m - \mu)}{2kv_m} + \dfrac{\sqrt{\rho_m\left(4Ckv_m + \rho_m(kv_m - \mu)^2\right)}}{2kv_m} \cong 120 \\[2ex]
\dfrac{\rho_m(kv_m - \mu)}{2kv_m} - \dfrac{\sqrt{\rho_m\left(4Ckv_m + \rho_m(kv_m - \mu)^2\right)}}{2kv_m} \cong 20.
\end{cases}
\tag{29}
$$

Solving system (29) yields,

$$
C = \frac{-2400kv_m}{\rho_m}, \quad \mu = \frac{kv_m(\rho_m - 140)}{\rho_m}.
\tag{30}
$$

Now, according to the national speed limit in Malaysia the maximum urban speed is $v_m = 60$ km/h. Furthermore, since the main objective of this research is to explore the effects of the fractional order $\alpha$, we set the free parameters as follows,

$$
k = 0.3, \ \beta = 2 \text{ and } \delta = 20.
\tag{31}
$$

Substituting all of the abovementioned parameters in Equation (30) we obtain,

$$
\rho = 70 + 50\tanh\left(4.16667 \cdot \left(0.3\frac{\Gamma(3 - \alpha)}{\alpha}x^\alpha + 3t - \lambda\right)\right).
\tag{32}
$$

Evidently, $\frac{\partial^\alpha \rho}{\partial x^\alpha} > 0$, $\forall\, x, t \in \mathbb{R}^+$, since $\frac{\partial^\alpha \tanh(x)}{\partial x^\alpha} > 0$ see [32], which verifies that the total flow $Q_{total}$ in Equation (17) is always positive.

In an effort to further understand the dynamic behaviors of the moving jam phase, we monitor the behavior of the middle wave point, namely $\rho_{Middle} = \frac{\rho_l + \rho_r}{2} = 70$. Substituting $\rho_{Middle} = 70$ in Equation (32) gives us the location and speed of the middle jam wave as,

$$
x\,|_{Middle} = 0.3^{\frac{-1}{\alpha}} \sqrt[\alpha]{\frac{\alpha(\lambda - 3t)}{\Gamma(3 - \alpha)}},
\tag{33}
$$

and the speed $S$ is,

$$
S = \frac{dx\,|_{Middle}}{dt} = -\frac{3 \cdot 0.3^{-1/\alpha}}{\alpha(\lambda - 3t)} \sqrt[\alpha]{\frac{\alpha(\lambda - 3t)}{\Gamma(3 - \alpha)}},
\tag{34}
$$

As a matter of fact, the fractional *GFFD* is defined as $x \in [0, +\infty)$, therefore, time is defined as $t \in [0, \lambda/3)$. Moreover, the speed in Equation (34) is always negative, indicating that the jam wave is moving upstream. Evidently, the moving jam speeds and locations will affect a traffic engineer's decision-making process when constructing transportation infrastructure such as signals and junctions. Indeed, transportation engineers are required to specify the value of fractional order $\alpha$ for each signal and junction, this value might be identified by collecting large data sets during the rush hours within several days.

Example. A trial of traffic light construction is simulated. Suppose a traffic engineer wants to construct a new traffic light at a short distance of 300 m before a fixed signal which is located at $x = 40$ km, bearing in mind that the resultant moving jam of the fixed signal

is not allowed to reach the new constructed signal. For the sake of comparison, we shall assume that the flow into the fixed traffic signal behaves under various fractional order values at $\alpha = 0.85, 0.90$ and $1.00$. Furthermore, the fixed traffic signal turns red, yellow and green for 72, 3 and 72 sec, respectively.

Now, based on Equations (33) and (34), $\lambda$ plays a key role in determining the location and speed of the moving jam phase. To that end, we use the initial state at $t = 0$ h to denote the start of the jam wave, and in particular the transition from the green to the red light through the yellow light. Taking into consideration that all of the jam waves must start forming at the same location from which the yellow light is flashing and the red light is about to turn on, as shown in Figure 3a.

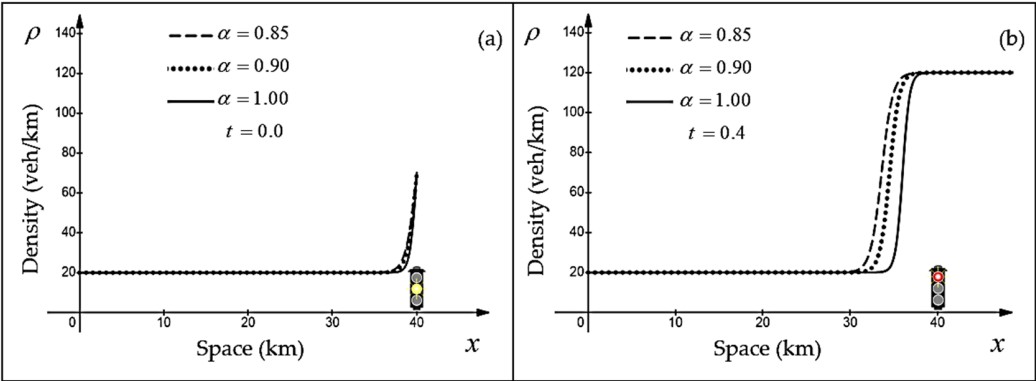

**Figure 3.** The density profiles of the solution of Equation (30) in the $\rho - x$ plane for various values of the fractional order: (**a**) The jamming formation as the yellow light is on at $t = 0$ h; (**b**) different moving jam waves at $t = 0.02$ h.

Using Equation (33), we obtain Table 1.

**Table 1.** Values of $\lambda$ correspond to various values of $\alpha$ order.

| $(x \mid_{Middle} = 40$ km, $t = 0$ h) | $\alpha = 0.85$ | $\alpha = 0.90$ | $\alpha = 1.0$ |
|---|---|---|---|
| | $\lambda_{0.85} = 8.710$ | $\lambda_{0.90} = 9.648$ | $\lambda_1 = 12$ |

Now, benefiting from Table 1 and Equations (33) and (34) the traffic engineer can specify the speeds and locations of various moving jam waves forming as the traffic signal turns red for 72 s, as illustrated in Table 2.

**Table 2.** Locations (km) and moving jam speeds (km/h) correspond to various values of the $\alpha$ order.

| | $\alpha = 0.85$ | $\alpha = 0.90$ | $\alpha = 1.00$ |
|---|---|---|---|
| $t = 0.02$ h | $x_{\alpha=0.85} = 39.672$ | $x_{\alpha=0.9} = 39.720$ | $x_{\alpha=1} = 39.80$ |
| | $S_{\alpha=0.85} = -15.788$ | $S_{\alpha=0.9} = -13.616$ | $S_{\alpha=1} = -10.00$ |

Therefore, when the fixed traffic signal turns red, the middle jam waves reach the locations $x_{\alpha=0.85} = 39.672$, $x_{\alpha=0.9} = 39.720$ and $x_{\alpha=1} = 39.80$ associated with different fractional order values $\alpha = 0.85$, $\alpha = 0.90$ and $\alpha = 1.00$, respectively.

Now, if the traffic engineer planned to set a traffic signal 300 m before the fixed signal, then he has three scenarios according to the value of the fractional order $\alpha$, as depicted in Figure 4. Consequently, if the flow of the fixed signal follows the $\alpha = 0.85$ type, then the moving jam will reach the location $x = 39.700$ km even before the fixed traffic signal turns green and, as a result, the proposed location is inadmissible. However, if the flow obeys the $\alpha = 0.90$ type, then the moving jam will reach 280 m prior to the fixed signal. Thereupon, the new signal might be constructed at the proposed location. In the same manner, the last scenario when $\alpha = 1.00$ is accepted too.

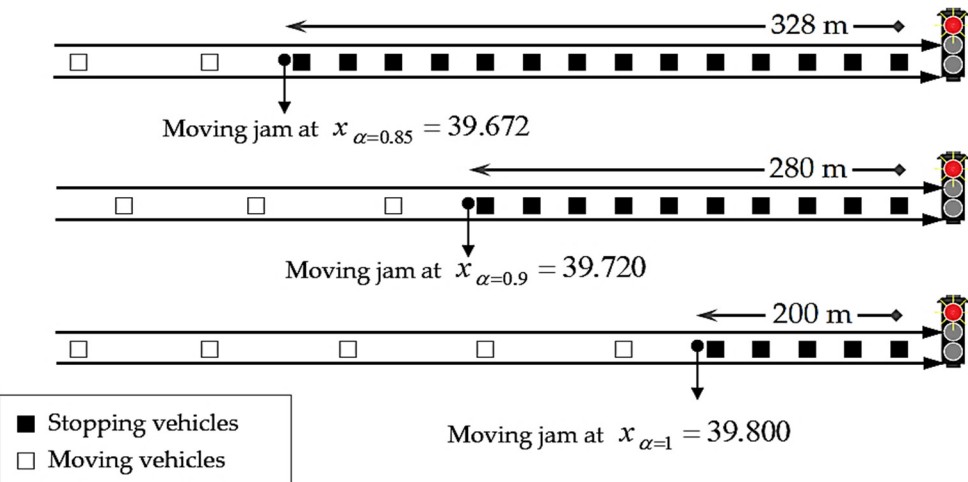

**Figure 4.** Three expected scenarios of the moving jam locations for various values of the fractional order $\alpha$.

Furthermore, as illustrated in Table 2 and Figure 4, the fractional order plays a crucial role in explaining the heterogeneity of the human response to incoming information on roads. In fact, the fractional order seems an important indicator of the driving styles. For example, drivers who follow the $\alpha = 0.90$ model admit moving jam speed at $S_{\alpha=0.9} = -13.616$, meanwhile those who follow the $\alpha = 0.85$ type show a more aggressive driving style, inasmuch as their moving jam speed is $S_{\alpha=0.85} = -15.788$. Hence, drivers in the proposed fractional model react differently based on the fractional order value and can deviate from the equilibrium model at $\alpha = 1$. To conclude, the smaller the fractional order, the faster the driving style, moreover the farther away drivers are from the red-light traffic signal.

So far, however, there has been very little discussion about the fractional LWR so more research is required to better understand the impacts of the fractional order. Recently, very few authors have been interested in the fractional vehicular model. Kumar et al. [46] solved the fractional LWR with the aid of the LFHPSTM and the LFRDTM methods. The authors used the initial and boundary conditions to reach a family of hyperbolic solutions based on the Mittag-Leffler function. Yet, there are two limitations of their study: firstly, the model did not involve the dispersion term, which reflects that vehicles will not diffuse from or out of the surroundings; second, vehicles are supposed to move in constant speed which is extremely rare in real world scenarios. Furthermore, they did not provide a physical meaning of their solutions. Yang Li et al. [47] used the LFLVIM method, which is the coupling method of local fractional variational iteration method and the Laplace transform, to solve the boundary value problems. Unlike our solutions, their results were non-differentiable approximate solutions which admit discontinuity and a negative density profile. Meanwhile, Wang et al. [48] studied the Cauchy problem of fractal dynamical models of vehicular traffic flow within the local fractional conservation laws based on the local fractional surface integral, the authors derived linear and nonlinear fractal differential equations for the LWR model. Still, the model can fit both constant and nonconstant velocity which produces a non-linear model. The authors, however, did not provide a solution of their model. Additionally, Jassim [49] found approximate solutions for partial differential equations that occur in fractal vehicular traffic using the LFLDM and the LFSEM methods. He agreed that these approaches provide us with a simple way to compare the approximate solution with less calculation compared with the local iteration method. Using boundary value problems for linear partial differential equations resulting from fractal vehicular traffic flow, the approximate solution of the problem was successfully achieved. However, the author only approached the linear LWR model, and offers no physical interpretations. Singh et al. [50] verified the existence and uniqueness of solutions for local fractional differential equations arising in fractal vehicular traffic flow. The authors successfully solved the model by using the basic method in cases of non-linear

non-homogeneous differential equation. However, the authors modified the initial value to verify the convergence of the solutions, which are still non-differentiable. They claimed that the proposed method offers less errors and computations and can be effectively used in fractal vehicular traffic flow models. In conclusion, the abovementioned papers are the closest research that has been conducted that refers to the fractional LWR and which are considered valuable and productive. However, we would highlight two disadvantages. First, the dispersion term was absent indicating, that those models did not consider the fractional Fick's law of diffusion which certainly leads to non-physical results on roads such as drivers deliberately adjusting their speed when jamming. Second, many physical phenomena could not be expected i.e., the anomalous behavior.

From separate schools of thought, Raissi et al. [51] introduced a new approach using physics-informed neural networks in which two different problems were outlined, namely a data-driven solution and a data-driven discovery. They developed two unique types of algorithms: the continuous time and the discrete time models. However, the geometric shape of the moving jam solution acts similarly to the obtained results in our study. Patently, the major advantage of their work is that the constructed model is not restricted to the Greenshield density–speed relation, accordingly this model is of huge relevance to future research into this area.

## 6. Conclusions

In this study, we proved that uphill dispersion addresses wrong-way travel so that the proposed model produces only anisotropic traffic behavior. However, the most obvious finding to emerge from this study is that the fractional order plays a significant role in describing certain features of traffic behavior in which each signal, junction or even road has its own driving style: aggressive or timid. Contrary to the classical calculus where the LWR model has only one state at $\alpha = 1$, the proposed fractional model produces infinitely as many states as $\alpha \in (0, 1]$ such that each different value has its own characteristics and implications, as a result the current model reveals various moving jam speeds and locations since the model exhibits anomalous behavior. The implementation of the trial equation method is a very simple and direct way of solving diverse categories of the fractional differential equations. Eventually, we believe that the study of the fractional uphill model in the paper will help traffic engineers to perfectly develop and improve the transportation infrastructure. For further research, we suggest further study of the time fractional model and the discovery of its consequences.

**Author Contributions:** Original draft preparation, investigation, conceptualization, analysis R.M.S.; review, editing and supervision S.S.J. All authors have read and agreed to the published version of the manuscript.

**Funding:** This research received no external funding.

**Data Availability Statement:** Not applicable.

**Conflicts of Interest:** The authors declare no conflict of interest.

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
