# Peer review of "A Space Fractional Uphill Dispersion in Traffic Flow Model with Solutions by the Trial Equation Method"

_infrastructures, doi:10.3390/infrastructures8030045_

Round 1

Reviewer 1 Report

The author considers the A Space Fractional Uphill Dispersion in Traffic Flow Model 2 with Solutions by the Trial Equation Method. The paper contains some interesting results but a major revision is required.

1.There are no good use of punctuation marks in the paper, especially in your equations. Equations are statements too and so should be punctuated. 

2. The authors stated that "Eq. (21) represents the fractional uphill model, as  approaches 1, the model tends to 173 equilibrium values" This must be illustrated in (1)

3. 

Many of the results and conclusions of this paper are quite basic. I recommend expanding: The introduction, Conclusions and Results sections. The aim should be to: (a) give a broader view of the literature on the topic and the current state-of-the-art; (b) clarify and discuss the novelty and the significance of the results obtained here, and compare them with those available in the literature, also including discussions on potential applications. I cannot support publication unless the authors fully undertake all the above actions.

The following references should be incorporated to increase the readability of the manuscript:

DOI:10.32604/cmc.2020.012314

https://doi.org/10.1140/epjp/s13360-020-00819-5 https://doi.org/10.1016/j.chaos.2020.109628 https://doi.org/10.1016/j.chaos.2018.09.043

Author Response

Attached in PDF our response. Please consider.

Reviewer 2 Report

After thoroughly reviewing the paper, I have identified several areas where major improvements are needed.

Firstly, the results presented in the paper are not realistic and do not accurately reflect real-world conditions. For example, in Figure 3, the shockwave speed is too fast and lasts for more than 10 kilometers in all cases, which is not consistent with observations in urban areas. Additionally, the maximum density of 200 vehicles per kilometer with an average vehicle length of 5 meters does not take into account the necessary space between vehicles for safety and traffic flow.

Secondly, the paper does not sufficiently explain the motivation for using a fractional derivative and how it improves upon the results obtained from the LWR model. Additionally, recent research has shown that using physics-informed neural networks can replace the use of fractional differential equations in the proposed model and it's important to compare the proposed model with these recent techniques.

Lastly, the paper also needs to be properly cited and give credit to the authors whose work was used. The section 2 is majorly inherited from literature, however, it's not properly cited.

I believe that with these major improvements, the paper will be a valuable contribution to the field of traffic flow modeling. I would appreciate it if the authors could address these issues and revise the manuscript accordingly before it is considered for publication. I would be happy to provide further feedback or answer any questions you may have.

Thank you for considering my comments.

Author Response

(The authors gave the same response as above.)

Reviewer 3 Report

This paper uses a space fractional uphill dispersion in traffic flow model to address the wrong way travel and to describe the anomalous transport behavior with considering various driving styles. Overall the structure of the paper is clear, and methodology is well explained. I have a few comments:

  • Some of the related literature are not properly cited. Please update the references.

  • Please add units for the axises in figure 1 and 2.

Author Response

(The authors gave the same response as above.)

Round 2

Reviewer 1 Report

The author(s) have attended to all the issues raised satisfactorily. I therefore recommend the paper form publication

Author Response

Dear Professor,

Hope you are doing well.

Your effort is highly appreciated.

Thanks for guiding and helping us to greatly improve the quality of the manuscript.

Kind regards,

Rfaat Moner Soliby                                                         

Reviewer 2 Report

The detailed comments are ignored by the authors. The text is appended below . However, please see the attached file for detailed comments. 

The paper has two main objectives: to modify the traffic flow model by introducing uphill dispersion to recover wrong-way travel and eliminate advected discontinuity. Secondly, to describe anomalous transport behavior by fractalizing the model to include fractional dynamics in space and using GFFD fractional derivative. The paper concludes by applying the trail equation method and simulating solutions for specific cases to help transportation engineers understand traffic behavior and make informed decisions about traffic signal networks.

The following are major comments

1.       The proposed method aims to improve traffic flow by addressing issues that lead to congestion. However, it is not clear how it would specifically benefit urban transport managers who are already using advanced technologies to manage traffic. The authors should provide more specific information on how this study can help urban transport managers in their day-to-day operations.

2.       It would be beneficial to dedicate a paragraph specifically discussing the advantages of the proposed approach over existing models. Additionally, it would be informative to compare the proposed model with at least one of the LWR, Pyne-Whitham, or Aw-RZ models through numerical experiments in the results section. This would provide a clear understanding of the strengths of the proposed method and how it compares to existing models in terms of accuracy and performance.

3.       How much time is required to solve this model? Is it possible to solve it in real time with the signal timings included?

4.       Recheck all the references. Reference titles are missing e.g., [11],[15], [27], etc.

5.       Conclusions should be in section 6. (Line 71, P2)

6.       It appears that a significant portion of the information in Section 2 is taken from previous literature. It is important to properly cite the sources and give credit to the authors whose work was used. Additionally, it would be helpful to clearly state and explain the specific contributions made by the current study in relation to the literature.

7.       Traffic engineers may not be familiar with advanced mathematics and some of the equations used in the model may appear complex and difficult to understand. To make it more accessible, it is recommended to include detailed steps and explanations in an appendix, starting from the basic equations and progressing to the final results. This will make it easier for traffic engineers to understand and use the model.

8.       P4, Line3. It is unclear why drivers would make sudden changes when there is a timed warning signal phase (yellow light). It is suggested to further explain this phenomenon in the manuscript and consider revising the manuscript to provide a clearer understanding of this behavior.

9.       With regards to uphill diffusion, the dispersive flow may not be negative altogether as there is still flow downstream and vehicles are actually moving towards areas of low concentration. It could be beneficial to include Q_dispersive = -D (∂p/∂x) - K (∂p/∂t) instead of making the whole term positive in the total flow. This would take into account the negative dispersive flow and provide a more accurate representation of the system. It is recommended to explain this reasoning in the manuscript and revise it accordingly.

10.   Please cite the active transport term on P5, Line 146.

11.   Explain free parameters on P 7 Line 207. How do you set these values? I would like to see a sensitivity analysis.

12.   The maximum density of 200 vehicles per kilometer with an average vehicle length of 5 meters does not provide an accurate representation of reality as it does not take into account the necessary space between vehicles for safety and traffic flow. A more realistic representation would include a minimum headway distance between vehicles to ensure safe and efficient traffic flow. It is suggested to revise this aspect of the model to ensure its accuracy.

13.  Explain the lambda term and how it is related to jam movement.

14.   The results presented in Figure 3 may not be realistic as the shockwave speed appears to be too fast and is lasting for more than 10 kilometers in all cases. This is not consistent with what is observed in real-world urban areas where there are multiple traffic signals. It is suggested to revisit the assumptions and parameters used in the model and re-evaluate the results to ensure their realism and relevance for an urban area. Additionally, it is important to consider the effect of traffic signals on the shockwave and its speed. This could be done by either simulating the effect of traffic signals on the shockwave or by comparing the results with real-world observations.

15.   The results for the case of a=1 seem more reasonable as compared to other values. This is because when a=1, it corresponds to the integer derivative or the LWR model which is a well-established model for traffic flow. It is unclear why a fractional derivative is being applied when the results for the LWR model are already reasonable. It could be beneficial to explain the motivation for using a fractional derivative and how it improves upon the results obtained from the LWR model in the manuscript.

16.   I would like to see the simulation and the data related to it.

17.   Recently, researchers have been using physics-informed neural networks, which involve ordinary partial differential equations with residual points from observations, to predict traffic states. It would be valuable to explain how the use of fractional differential equations with neural networks. A dedicated paragraph in the literature review section would be useful for this purpose, and it's important to cite relevant studies that have used physics-informed neural networks for traffic state prediction.

1.       Shi, R.; Mo, Z.; Di, X. Physics-Informed Deep Learning for Traffic State Estimation: A Hybrid Paradigm Informed by Second-Order Traffic Models. In Proceedings of the AAAI Conference on Artificial Intelligence, Online, 2–9 February 2021; Volume 35, pp. 540–547. 

2.       Usama, M.; Ma, R.; Hart, J.; Wojcik, M. Physics-Informed Neural Networks (PINNs)-Based Traffic State Estimation: An Application to Traffic Network. Algorithms 2022, 15, 447. https://doi.org/10.3390/a15120447

Author Response

Dear Professor,

Hope you are doing well.

Your effort is highly appreciated.

Attached as PDF. please consider.

Kind Regards

Rfaat Moner Soliby

Round 3

Reviewer 2 Report

Before publication, the following issues need to be addressed:

  1. Your response to comment 3 is unclear. The commenter asked about the problem solution time (in seconds) using moderate computational resources, to determine if the solution approach is efficient enough for real-time implementation. Please clarify your response.
  2. Your response to comment 7 is insufficient. Transportation professionals are concerned that your model's solution remains a black box. They are familiar with the solutions to ordinary differential equations and wonder why your approach, which has a black-box solution even for a first-order differential model, is necessary. Please address this issue more thoroughly.
  3. Comments 9 and 11 should be addressed in the current model rather than left for future research, as they are integral to the current model.
  4. Please provide the data and code needed to reproduce your results for review purposes only, as requested in comment 16.
  5. The paragraph on physics-informed related studies does not include all the necessary information related to traffic state estimation as desired in comment 17.
  6. Please include all explanations in the revised manuscript.

Author Response

Please find attached our response in PDF.
